# Visual Detection of Malaria Parasite-Parasitized Erythroblasts in Peripheral Blood via Immunization-Based Model

**DOI:** 10.3390/vaccines13090988

**Published:** 2025-09-21

**Authors:** Kumpei Ito, Yuki S. Tateishi, Takashi Imai, Shinya Miyazaki, Yukiko Miyazaki, Wataru Kagaya, Mai Nakashima, Miho Sase, Misato Yoshioka-Takeda, Chikako Shimokawa, Kyoko Hayashi, Kentaro Itokawa, Osamu Komagata, Ha Ngo-Thanh, Aoi Shimo, Tamasa Araki, Takeshi Annoura, Takashi Murakami, Hajime Hisaeda

**Affiliations:** 1Department of Parasitology, National Institute of Infectious Diseases (NIID), Japan Institute for Health Security (JIHS), Tokyo 162-0052, Japanytateisi@niid.go.jp (Y.S.T.); tamasa@niid.go.jp (T.A.);; 2Department of Infectious Diseases and Host Defense, Gunma University, Maebashi 371-8511, Japan; 3Department of Microbiology, Saitama Medical University, Saitama 350-0495, Japan; 4Leprosy Research Center, National Institute of Infectious Diseases (NIID), Japan Institute for Health Security (JIHS), Tokyo 189-0002, Japan; 5Antimicrobial Resistance Research Center, National Institute of Infectious Diseases (NIID), Japan Institute for Health Security (JIHS), Tokyo 189-0002, Japan; 6Department of Protozoology, Institute of Tropical Medicine, Nagasaki University, Nagasaki 852-8523, Japan; 7Department of Ecoepidemiology, Institute of Tropical Medicine, Nagasaki University, Nagasaki 852-8523, Japan; 8Department of Molecular Protozoology, Research Institute for Microbial Diseases (RIMD), Osaka University, Osaka 565-0871, Japan; 9Department of Tropical Medicine and Malaria, Research Institute, National Center for Global Health and Medicine (NCGM), Japan Institute for Health Security (JIHS),Tokyo 162-8655, Japan; 10Department of Medical Entomology, National Institute of Infectious Diseases (NIID), Japan Institute for Health Security (JIHS), Tokyo 162-8640, Japan; 11National Hospital for Tropical Disease, Hanoi 10000, Vietnam; 12Research Center for Biosafety, Laboratory Animal and Pathogen Bank, National Institute of Infectious Diseases (NIID), Japan Institute for Health Security (JIHS), Tokyo 162-8640, Japan

**Keywords:** malaria, erythroblast, Giemsa staining, vaccine model, *Plasmodium berghei*

## Abstract

Background: Erythroblasts have recently been identified as host cells for malarial parasites, revealing a previously underappreciated host–parasite interaction. However, their extremely low abundance in peripheral blood has hindered progress, especially in elucidating the biological significance of parasitized erythroblasts (pEBs) *in vivo*. Methods: Here, we visualized pEBs in a murine model and established a method to increase their number in peripheral blood by immunizing mice with live *Plasmodium yoelii* 17XNL, followed by challenge with *P. berghei* ANKA. Results: Immunized mice were protected from cerebral malaria and survived longer, during which pEBs appeared in circulation and were detected using Giemsa-stained smears. All blood-stage parasite forms were identified within pEBs, including enucleating erythroblasts. Conclusions: This model enables *in vivo*/*ex vivo* analysis of pEB biology without bone marrow/spleen isolation, thus lowering technical/ethical barriers for the field.

## 1. Introduction

More than a century ago, Laveran discovered the invasion of red blood cells (RBCs) by malarial parasites, marking a milestone in malaria research [1]. Recently, our group and others have demonstrated that malaria parasites also invade erythroblasts [2,3,4,5,6,7,8,9], cells that differ significantly from mature RBCs owing to the presence of nuclei and organelles and some other surface markers [10]. Notably, parasite-parasitized erythroblasts (pEBs) express CD44 and MHC class I molecules [4], which play a key role in self/non-self-recognition by the immune system [11]. Such immune recognition involves T cell receptor–MHC class I interaction and FasL-Fas signaling, leading to phosphatidylserine externalization on the pEBs [12], a hallmark of apoptosis-like cells.

Despite the potential importance of pEBs in malaria pathogenesis and immunity—including as targets of CD8^+^ T cell responses relevant to vaccine design—progress in this field has been hindered by major technical and ethical barriers. One significant difficulty in studying pEBs is the rarity of these cells in peripheral blood. In humans, pEBs have not been observed in the peripheral blood or even in murine malaria models, and in the last 20 years of our research, only a few times have pEBs been identified in peripheral blood smears from PyNL-infected mice. Owing to this rarity, studying pEBs *in vivo* or *ex vivo* in mice requires sacrificing the animals to dissect and analyze the spleen or bone marrow [4,5,7,12]. In humans, splenic extramedullary hematopoiesis is primarily observed during the fetal stage, but it has also been reported in sporadic cases during adulthood, such as in patients with bone marrow failure [13] and in individuals with sickle cell disease [14]. In contrast, in adult mice, the spleen continues to serve as an active site of extramedullary hematopoiesis under physiological conditions. Although pEBs have been detected in human bone marrow samples and *in vitro* culture systems [2,3,6,9], their use in *Plasmodium falciparum* infection studies depends on maintenance procedures based on stem cell differentiation protocols [15] and the availability of human-derived materials such as RBCs and serum [16]. Moreover, due to MHC restriction, functional analysis of T cell–pEB interactions is generally not feasible in human *in vitro* experiments, though some insights have been gained by using rodent malaria model *ex vivo* experiments [4,12]. Clinically, erythroblasts are typically confined to the bone marrow, and obtaining such samples poses ethical and logistical challenges. Consequently, pEBs have not been reported in peripheral blood.

To overcome these challenges, we employed a murine immunization–challenge model using non-lethal *P. yoelii yoelii* 17XNL (PyNL) followed by *P. berghei* ANKA (PbA) challenge. Strikingly, we found that this approach allowed for consistent detection of pEBs in the peripheral blood at high parasitemia (>60%) using standard Giemsa-stained smears—eliminating the need for bone marrow/spleen isolation or transgenic parasites. This model not only facilitates *in vivo* or *ex vivo* study of pEB biology but also opens a new experimental window to investigate immune responses and parasite development in this previously hidden niche.

## 2. Materials and Methods

### 2.1. Mice

Male and female C57BL/6 mice (6 to 8 weeks old) were obtained from SLC (Shizuoka, Japan). Upon arrival, the animals underwent a one-week acclimatization period and were subsequently housed under specific pathogen-free (SPF) conditions at the animal facility at the Department of Infectious Diseases and Host Defense, Gunma University, Department of Microbiology, Saitama Medical University and NIID, JIHS.

Mice were kept in a controlled environment at 25 °C with a 12-h light/dark cycle and were provided with unrestricted access to food and water. Male and female mice were used, and no sex-related differences were observed.

All experiments using mice were performed in accordance with our institutional guidelines for the use of laboratory animals and approved by the Review Board for Animal Experiments of Gunma University, Saitama Medical University and National Institute of Infectious Diseases (approval ID:19-016, 3322, 123046).

### 2.2. Malaria Parasites and Blood-Stage Infection

Blood-stage rodent malaria parasites used in this study included PyNL, *P. yoelii yoelii* 17XL (PyL), and PbA. These strains were generously provided by Professor M. Torii (Ehime University, Japan) and Dr. K. Suzue (Gunma University, Maebashi, Japan). The PyNL was established by Landau and Killick-Kendrick [17] and was subsequently transferred through Cox’s laboratory [18]. In 1984, it was introduced into Japan by Dr. Waki (Gunma University) from the Playfair laboratory [19] at Middlesex Hospital Medical School. The PyL strain is a variant of PyNL that emerged during serial passage [19]. In 1984, it was introduced into Japan by Dr. Waki (Gunma University) from the Playfair laboratory. The PbA was isolated from *Anopheles dureni millecampsi* mosquitoes from the Democratic Republic of Congo by Vincke and Bafort [20]. It was also introduced into Japan in 1984 by Dr. Waki (Gunma University) from the London School of Hygiene and Tropical Medicine.

Parasite stocks were cryopreserved in liquid nitrogen and maintained by serial passage through donor wild-type mice via intraperitoneal (i.p.) injection. Procedure in donor mice: Thawed cryopreserved blood containing parasitized red blood cells (pRBCs) (0.1–0.5 mL) was injected i.p. into donor mice. Two to four days later, mice were anesthetized with isoflurane, and several drops of blood were collected from the tail tip into a tube containing 1 mL of RPMI 1640 medium (Sigma-Aldrich, St. Louis, MO, USA). For experimental infections, mice were injected i.p. with 25,000 PyNL-pRBCs, 25,000 PyL pRBCs, or 50,000 PbA pRBCs suspended in 0.5 mL RPMI 1640 medium. The parasitemia of donor mice at the time of passage was maintained at 0.5–1%.

Parasitemia was determined from thin blood smears prepared at the time of sampling. Each smear was independently examined by more than one investigator, and additional smears were prepared if the blood condition compromised slide quality. Parasitemia was determined using methanol-fixed smears stained with 3% Giemsa solution (Sigma-Aldrich, St. Louis, MO, USA; or Nacalai Tesque, Kyoto, Japan). For each sample, at least 2000 RBCs were counted. Measurements were performed at specific time points depending on the parasite strain: days 8, 14, 18, 20, 23, 27, and 35 after PyNL infection; days 4, 6, 8, 12, 19, 24, 32, 40, and 52 after PbA infection; and days 6, 8, and 10 after PyL infection.

### 2.3. Immunization (Live Vaccination) and Challenge Infection

For immunization, mice received 25,000 pRBCs of live PyNL by intraperitoneal injection. Forty days after immunization, mice were challenged with 50,000 pRBCs of PbA. Control non-immunized mice were injected with PBS at the same time. Parasitemia levels was monitored, and survival was monitored by counting pRBCs in Giemsa-stained blood smears of tail blood. The infected mice were monitored daily for survival over the course of infection by direct observation.

### 2.4. Microscopic Examination of pEB

At 12 and 24 days after PbA challenge infection, blood smears were prepared from 21 PyNL-immunized and subsequently PbA-challenged mice. The thin blood smears prepared on microscope slides were fixed with methanol for 30 sec and stained with 3% Giemsa solution for 30 min. After staining, the slides were washed with tap water and examined under a microscope to detect pEBs, the same as pRBC detection. For each specimen, parasitemia (%) and the ratios of pEBs to pRBCs or to total EBs were determined by analyzing a minimum of 10 and a maximum of 700 microscopic fields (1000× magnification, 100× objective lens with immersion oil).

Microscopic examinations were independently performed by at least eight researchers to identify pEBs, and representative photographs were taken using a Nikon ECLIPSE Ci (Melville, NY, USA) and NIS-Elements L software (https://www.microscope.healthcare.nikon.com/en_AOM/products/cameras/nis-elements-l, access on 18 September 2025). We selected optimal fields on the smears where 100 to 400 non-overlapping cells could be observed. In our smears, low parasitemia typically yielded 300–400 RBCs per field, whereas high parasitemia due to anemia resulted in 100–200 RBCs per field.

We counted at least 200 pRBCs to ensure reliable detection of pEBs. For example, at 0.1% parasitemia, approximately 600 fields needed to be examined (400 RBC × 600 fields × 0.001 = 240 pRBC). Conversely, at high parasitemia, only about 10 fields were required (100 RBC × 10 fields × 0.9 = 900 pRBC). To more rigorously evaluate the presence of pEB in PyNL or PyL-infected mice, microscopic examination was continued until 10,000 to 40,000 pRBCs were counted. Quantitative data from three trained researchers were combined for each specimen to improve accuracy.

### 2.5. Evaluation of Blood–Brain Barrier

To evaluate blood–brain barrier (BBB) integrity, mice received an intravenous injection of 0.2 mL heparin solution, followed by 0.2 mL of 1% Evans Blue (Sigma-Aldrich, St. Louis, MO, USA) in PBS. One hour later, animals were euthanized, exsanguinated, and perfused intracardially with 20 mL of PBS. Brains were then collected and incubated in 2 mL of formamide for 48 h at 37 °C. The absorbance of Evans Blue was measured at an optical density (O.D.) of 630 nm. A standard curve, prepared from serial dilutions of the Evans Blue solution, was used to quantify the dye concentration.

### 2.6. Staining of Parasitized Cells and Fluorescence Microscopy

On day 20 post-challenge infection, 100 μL of blood was collected from PyNL-immunized followed by PbA-challenged mice. TER119-positive erythroid cells were isolated using anti-TER119 microbeads and an LS column from a MACS system (Miltenyi Biotec, Bergisch Gladbach, Germany) according to the manufacturer’s protocol, which includes an Fc-block (BioLegend). The isolated cells were then stained for surface markers with APC-conjugated anti-CD44 antibodies (BioLegend, San Diego, CA, USA). Following fixation and permeabilization with the Cyto-Fast™ Fix/Perm Buffer Set (BioLegend, San Diego, CA, USA), intracellular staining was performed using a FITC-conjugated anti-*P.berghei* HSP70 antibody (prepared in-house). Finally, the cells were stained with Hoechst 33342 (5 μg/mL; Sigma-Aldrich, St. Louis, MO, USA). Images were taken using a BZ-X810 fluorescence microscope (Keyence, Osaka, Japan)

### 2.7. Statistical Analysis

Significant differences in survival were tested with a Gehan–Breslow–Wilcoxon test. Data were analyzed with one-way ANOVA, and Mann–Whitney U test. GraphPad Prism version 8.0 (GraphPad Software Inc., San Diego, CA, USA) was used in the above analysis.

## 3. Results

### 3.1. PyNL Infection and Rare Detection of Parasitized Erythroblasts in Peripheral Blood

Parasitized erythroblasts (pEBs) were rarely detected in the peripheral blood of mice infected with PyNL (Figure 1A). More than 1500 Giemsa-stained blood smears were examined, and the frequency of pEBs relative to parasitized red blood cells (pRBCs) was extremely low. PyNL is a non-lethal, self-limiting strain; infected mice recovered within one month, and no relapse was observed up to 60 days after infection (Figure 1B).

### 3.2. Live PyNL Immunization Partial Protection Against PbA Challenge Infection in Mice

Immunization of mice with PyNL served as a live vaccination, followed by a challenge infection with PbA (Figure 2A,B). PbA infection in C57BL/6 mice is a well-established model of human cerebral malaria, referred to as experimental cerebral malaria (ECM) [21]. Typically, between days 7 and 10 post-infection with PbA, mice develop ECM even when parasitemia remains relatively low (Figure 2C), due to disruption of the blood–brain barrier (BBB, Figure 2D).

To evaluate the integrity of the BBB, Evans blue dye was injected at day 8 or 24 after PbA challenge (Figure 2A). In PbA-infected control non-immunized mice, ECM development caused BBB disruption, resulting in blue staining of the brain (Figure 2D). In contrast, no blue staining was observed in any of the PyNL-immunized and PbA-challenged mice, indicating effective protection against cerebral malaria. Even when these assays were performed at the time when PyNL-immunized and subsequently PbA-challenged mice were moribund, the integrity of the BBB was largely maintained, indicating that death in these mice was not caused by cerebral malaria. Moreover, these immunized mice showed no neurological abnormalities typically associated with ECM, such as ataxia, seizures, or paralysis. Thus, PyNL-immunized, challenged mice avoided ECM, maintained the BBB (Figure 2D), and showed prolonged survival compared with PbA-infected control non-immunized mice (Figure 2B).

However, 65% of the immunized challenged mice eventually succumbed to hyperparasitemia (60–90% parasitemia) and died. In contrast, 35% of the immunized mice survived for an extended period, maintaining low parasitemia (less than 1% even 40 days after PbA challenge) (Figure 2B,C).

### 3.3. Visual Detection and Increased Proportion of pEBs in Peripheral Blood

As described in the introduction and Figure 1, since erythroblasts are typically rare from peripheral blood, to overcome this limitation, we aimed to develop a method to increase the presence of pEBs in peripheral blood. Our previous studies demonstrated that immunization of mice with PyNL regulates the overactive immune response to PbA infection [22] and that infection-induced splenomegaly facilitates enhanced erythropoiesis [23]. Thus, PbA-challenged PyNL-immunized mice tended to maintain high parasitemia and survived longer, which was achieved through massive erythropoiesis; otherwise, the mice died quickly due to severe anemia. Based on these findings, we attempted to investigate whether PyNL-immunized mice could be utilized to analyze pEBs in peripheral blood. In the past 20 years of our research, we have examined more than 100 non-immunized mice around days 7 to 14, and PbA-pEBs have never been detected in peripheral blood smears. Remarkably, we observed significantly higher numbers of pEB in the peripheral blood of PyNL-immunized PbA-challenged mice with hyperparasitemia (Figure 2C). Under these conditions, pEBs were more readily identified through microscopic examination with Giemsa-stained blood smear slides (Figure 3A) compared to previous observations. This approach also revealed that PbA can invade erythroblasts, similar to PyNL. Furthermore, we observed rare instances of erythroblasts undergoing enucleation while harboring parasites (Figure 3A, right panel, Figure 5o), paralleling findings of *in vitro* studies of Pf [9].

To confirm that the detected parasites were PbA rather than PyNL, and that the identified pEBs were indeed erythroblasts rather than other cell types, we performed intracellular staining with a fluorescent antibody against *P. berghei* HSP70, together with surface staining for CD44 (pink), which is expressed on erythroblasts [4,24,25]. Fluorescence microscopy images are provided in Figure 3B. As additional evidence that the cells detected in peripheral blood were PbA-pEBs, we found that while pRBCs exhibited FITC (green) and nuclear (blue) staining, they did not express CD44, in contrast to pEBs.

### 3.4. Time-Course Analysis of Parasitemia and pEBs from Primary Infection with PyNL to Challenge Infection with PbA

As described in the earlier part of this study, a relatively large number of pEBs were detected in the peripheral blood of immunized, challenged mice. We next examined the changes in the number of pEBs. Blood smears were prepared on day 20 after PyNL infection and on days 12 and 24 after PbA challenge, and pEBs were counted (Figure 4A). During PyNL infection and subsequent challenge infection with PbA, parasitemia showed characteristic changes (Figure 4B). The peak parasitemia of the primary PyNL infection occurred around day 20. On day 20 after PyNL infection, pEBs were detected in one out of seven mice, with a frequency of 0.000246 pEB per pRBC (10,000–40,000 pRBCs examined). On day 12 after PbA challenge, no pEBs were detected in peripheral blood. By day 24 after PbA challenge, pEBs were detected in mice that developed high parasitemia (>60%), with a frequency of 0.0029 ± 0.0015 pEB per pRBC (Figure 4C). In contrast, in long-term surviving mice that mounted strong immunity and maintained parasitemia below 1% (Figure 4B), no pEBs were observed (Figure 4C). We summarize the data from 17 mice in Figure 4D (pEB/pRBC vs. % parasitemia) and Figure 4E (pEB/EB vs. % parasitemia).

Parasitized erythroblasts were most commonly detected after parasitemia had risen above 60% following challenge infection. However, as an exception, we also observed pEBs in a mouse with a parasitemia as low as 0.2%. In PyNL-immunized and PbA-challenged mice, the ratio of parasitized erythroblasts to parasitized erythrocytes reached at most 0.004 (0.4%) (Figure 4D). This nevertheless represents a significant advance, since in non-immunized mice, parasitized erythroblasts were hardly observed, even immediately prior to their death within two weeks.

Longitudinal observation of individual mice revealed that even animals in which pEBs were absent at 20% parasitemia later exhibited pEBs once parasitemia exceeded 60% (Figure 4C). This suggests that a parasitemia level around 60% may represent a threshold for the appearance of pEBs. In samples where pEBs were detected, the ratio of pEBs to total EBs was approximately 0.5 (Figure 4E).

### 3.5. Immunization Is Required for Increased Proportion of pEBs in Peripheral Blood

Here, a question arose: could the detection of pEBs simply depend on severe anemia caused by high parasitemia? Therefore, we used a different strain, PyL, which causes lethal anemia due to extremely high parasitemia [26]; many mice reached parasitemia levels over 80% within 10 days after infection (Figure 4G). We examined whether pEBs could be detected in PyL-infected mice on day 8 (Figure 4F). Even after carefully screening numerous microscopic fields (1000× magnification), each containing approximately 10,000–40,000 parasitized RBCs, we did not observe any pEBs (Figure 4H). These data suggest that even under conditions of severe anemia induced by heavy infection, the host requires additional time to mount sufficient hematopoiesis to supply RBCs. In contrast, PyNL immunization followed by challenge infection with PbA induces robust hematopoiesis, as reflected by marked splenomegaly. For example, the spleen weight on day 20 after PyNL immunization followed by PbA challenge reached 1–2 g, compared to only 0.05 g in naïve mice (data not shown).

### 3.6. Catalog of PbA-Parasitized Erythroblasts Under Microscopic Examination

Using Giemsa-stained smears, we documented all parasite stages within the erythroblasts, including merozoites, rings, trophozoites, schizonts, and gametocytes (Figure 2). Furthermore, we were able to capture a moment that appeared to show merozoites (daughter cells) being released from a schizont within a pEB (bottom panel of the schizont images in Figure 5). This represents a significant achievement in visualizing the complete developmental cycle of the non-transgenic parasite in peripheral blood erythroblasts without sacrificing the mice or collecting the bone marrow or spleen.

**Figure 5 vaccines-13-00988-f005:**
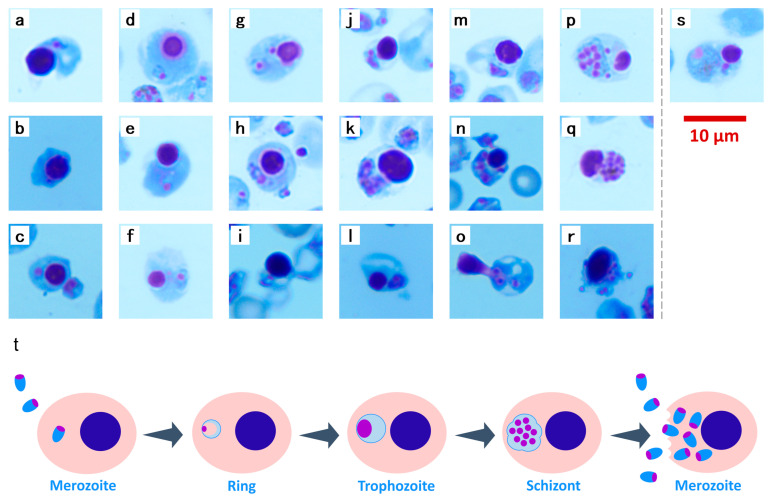
Catalog of *Plasmodium berghei* ANKA-parasitized erythroblasts (pEBs) under microscopic examination with Giemsa-stained peripheral blood smears. (**a**–**c**) Merozoite invasion, (**d**–**i**); ring, (**j**–**o**); trophozoite, (**p**–**r**); schizont. Stages from merozoite invasion to schizont development (left side to right side) and gametocyte (**s**) in pEBs were captured. Gametocytes are the sexual forms of the malaria parasite that develop in the mammalian host and are essential for transmission to mosquito stage. Scale bar: 10 μm. (**t**) A schematic diagram of pEBs and the developmental stages of the parasite within them. Red-purple indicates the nucleus of the *Plasmodium*, while sky-blue or cobalt-blue indicate its cytoplasm. Pale pink indicates the cytoplasm of the host erythroblast, and navy indicates the host nucleus. During maturation at the schizont stage, the host cell ruptures, releasing many merozoites (daughter cells) into the bloodstream, where they initiate a new invasion cycle of red blood cells or erythroblasts.

Erythroblasts are nucleated cells with a progressively condensing nucleus and cytoplasm that transition from basophilic (rich in ribosomes) to acidophilic (owing to hemoglobin accumulation) as they mature. They decrease in size in stages, from proerythroblasts with large nuclei and deeply basophilic cytoplasm to basophilic and polychromatic erythroblasts with mixed staining and, finally, to orthochromatic erythroblasts with pyknotic nuclei and acidophilic cytoplasm [27]. The ratio of PbA-parasitized polychromatic erythroblasts to orthochromatic erythroblasts in peripheral blood was 1:2.

## 4. Discussion

In this study, we introduce a novel methodology to reduce the barriers for future researchers to embark on studies on *Plasmodium*-pEB.

The detection of erythroblasts in the peripheral blood has important physiological implications. Under normal conditions, erythroblasts are retained within the bone marrow or spleen, and only mature erythrocytes circulate. The presence of erythroblasts in circulation may reflect stress erythropoiesis or altered bone marrow dynamics during malaria infection [28,29]. Infection with PyNL affects erythroblast differentiation, particularly the maturation of RBCs [30]. Consequently, immature erythrocytes accumulate in the spleen and bone marrow; however, they do not enter the peripheral circulation [30]. Based on our findings, we speculate that additional infection with PbA may influence the spleen and bone marrow in a way that allows immature erythrocytes to be released into the peripheral blood. Importantly, our observation that these circulating erythroblasts can be infected by *P. berghei* suggests that parasite replication is not confined to erythrocytes. Infection of these erythropoietic precursors may contribute to anemia by directly reducing the pool of developing RBCs and could facilitate parasite propagation within the host.

Additionally, circulating pEBs might interact with the host immune system differently than mature erythrocytes, potentially influencing immune recognition and clearance. In the PbA challenge following PyNL immunization, mice that eventually succumbed to infection exhibited a relatively high number of pEBs in their peripheral blood. Interestingly, these mice showed prolonged survival compared to the non-immunized controls, suggesting that the presence of pEBs may have had a transient impact on disease course. In contrast, PyNL-immunized mice that survived the PbA challenge did not show increased parasitemia, and no parasitized erythroblasts were observed in their circulation.

Although the present study cannot establish a causal link between circulating pEBs and protection against PbA challenge, our previous work demonstrated that CD8^+^ T cells recognize MHC class I antigens on parasite-infected erythroblasts, become activated, and release interferon-γ [4]. Furthermore, CD8^+^ T cells were shown to recognize Fas on infected erythroblasts, leading to phosphatidylserine exposure and subsequent macrophage-mediated clearance [12]. These findings suggest that circulating pEBs may provide additional substrates for immune recognition, thereby shaping host–parasite interactions during infection. Taken together, the data highlight the physiological relevance of circulating erythroblasts and raise the possibility that they could serve as both modulators of immune responses and potential indicators of disease progression in malaria.

Although pEBs have not been reported in human and rodent peripheral blood, this absence may reflect a historical lack of recognition rather than true absence. Since pEBs are rarely, if ever, described in standard parasitology textbooks, it is conceivable that such cells—if encountered—may be misidentified or overlooked under microscopy. By documenting the morphological characteristics of pEBs in peripheral blood (Figure 5), this study aims to provide a reference point for future observations. In the long term, automated detection using artificial intelligence and machine learning could further enhance the ability to recognize and quantify pEBs in blood samples.

Establishing *in vitro* culture systems for rodent malarial parasite pEB is critical for future studies. Although *P. yoelii* blood-stage parasites have successfully invaded RBCs *in vitro* [31], *P. berghei* can be maintained in short-term culture, allowing development from the ring to the schizont stage [32]; extending this to erythroblasts will require overcoming substantial technical hurdles.

The consistent detection of pEBs in peripheral blood using this PyNL immunization and PbA challenge model offers a novel *in vivo* and *ex vivo* system for studying erythroblast–parasite interactions without the need for invasive procedures or genetically modified parasites. This model provides a morphological reference that can support future studies aimed at understanding the functional role of pEBs in malaria pathogenesis and immunity.

The advantages of using pEBs from peripheral blood are outlined below. Unlike pEBs isolated from the spleen or bone marrow, which require mechanical or enzymatic dissociation of the tissue—procedures that may alter the original phenotype [33,34,35]—pEBs in peripheral blood are already dispersed. This facilitates experimental handling and increases the likelihood of isolating cells that closely retain their *in vivo* characteristics. Importantly, the identification of pEBs outside the bone marrow challenges the conventional view that such interactions are confined to hematopoietic tissues. Their presence in peripheral blood—previously considered negligible—may have been underestimated due to technical limitations or interpretive oversight.

While our findings contribute to the understanding of the biology and dynamics of pEBs in mouse models, there are several limitations that should be considered in the present study. One important limitation is that the detection of pEBs was mainly based on microscopic observation, and quantitative analysis using alternative methods was not performed. Therefore, further studies will be required to quantitatively assess pEBs using flow cytometry or other quantitative approaches. In addition, it is important to note that not all PyNL-immunized mice exhibited an increased proportion of pEBs in peripheral blood following PbA challenge. Accordingly, further studies will be necessary to establish immunization models in which pEBs consistently emerge more efficiently in peripheral blood.

Taken together, our findings suggest that erythroblasts circulating with parasites deserve renewed attention, both in experimental settings and in the clinical examination of malaria patients.

This system may facilitate the development of more accurate diagnostic techniques and improve our understanding of host–parasite dynamics. Ultimately, insights from such models could contribute to the refinement of cross-species vaccine strategies and the development of new therapeutic approaches.

## 5. Conclusions

This study demonstrates, for the first time, that parasite-parasitized erythroblasts (pEBs) can be consistently detected in the peripheral blood of mice using a PyNL immunization and PbA challenge model. This system enables non (mild)-invasive, *in vivo* observation without the need for transgenic parasites.

By establishing a reproducible method to visualize pEBs in peripheral blood, we provide a valuable experimental platform for future research into their role in malaria pathogenesis and host immunity. Our findings offer a foundation for developing improved diagnostic tools and informing novel strategies in malaria vaccine development.

## Figures and Tables

**Figure 1 vaccines-13-00988-f001:**
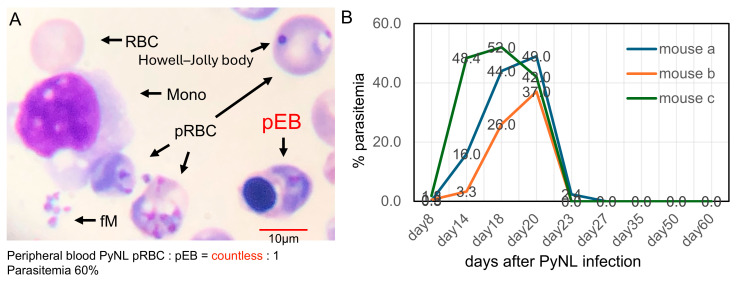
*Plasmodium yoelii* 17XNL infection and rare detection of parasitized erythroblasts in peripheral blood. (**A**) Parasitized erythroblasts (pEBs) were extremely rarely detected in Giemsa-stained peripheral blood smears from *Plasmodium yoelii* 17XNL (PyNL)-infected C57BL/6 (B6) mice (N > 1500 smears). The approximate ratio of PyNL-parasitized red blood cells (pRBCs) to pEBs was countless: 1. Monocytes (Mono) were observed attaching to RBCs and pRBCs, forming clusters. Free merozoites (fM), released from schizonts elsewhere, were also observed. The pRBC in the upper right contains both a ring-stage parasite and a Howell–Jolly body, a remnant of the host cell nucleus. Because it contains only nuclear debris and lacks an intact nucleus, this cell is not classified as an erythroblast. (**B**) PyNL, a non-lethal and self-limiting strain, resolved spontaneously in mice within one month after infection, and no relapse was observed up to 60 days post-infection. Results from three representative mice out of a total of 24 are shown.

**Figure 2 vaccines-13-00988-f002:**
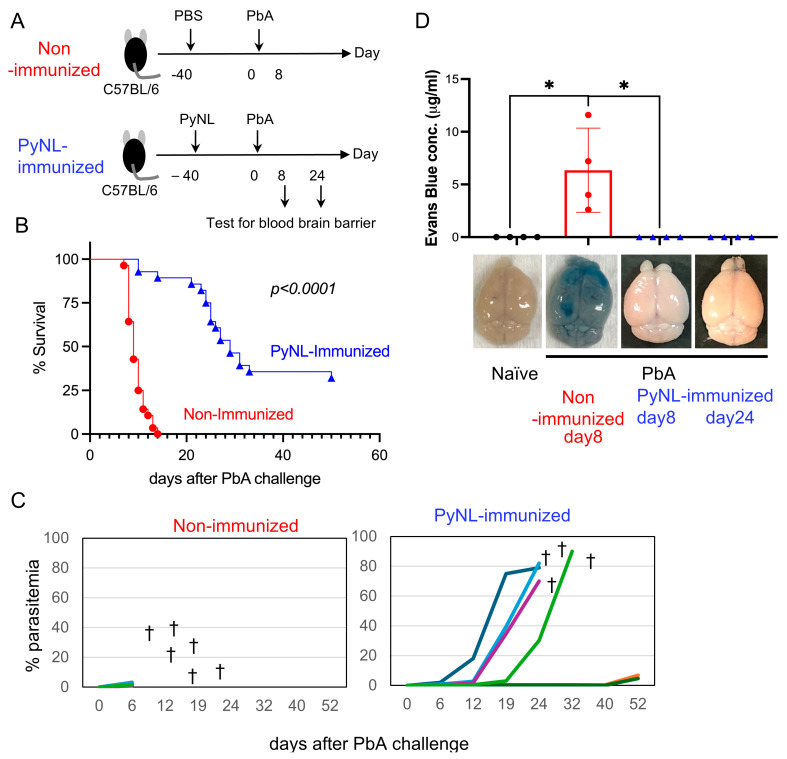
*Plasmodium yoelii* 17XNL (PyNL) infection provides partial protective immunity against *P. berghei* ANKA (PbA). (**A**) Mice infected with PyNL, which resolved within 30 days, were subsequently challenged with PbA. PyNL-immunized mice survived longer than control non-immunized PbA-infected B6 mice, which developed hyper-parasitemia. (**B**) Survival curves of non-immunized (red; N = 28) and PyNL-immunized (blue; N = 28) mice following PbA challenge infection. Data were pooled from five independent experiments. *p*-values were calculated using the Gehan–Breslow–Wilcoxon test. (**C**) Parasitemia of non-immunized (N = 6) and PyNL-immunized (N = 6) mice challenged with PbA. Each line with a different color represents data from an individual mouse. Data from one representative experiment out of five independent experiments are shown. † indicates death. (**D**) Assessment of blood–brain barrier integrity. Evans blue dye was injected intravenously, mice were perfused with PBS, and brains were collected and soaked in formamide. The concentration of the eluted dye was measured (N = 4 per group). Data were analyzed using one-way ANOVA [F (3,12) = 10.11, *p* = 0.0013] and the Mann–Whitney U test. * *p* < 0.05.

**Figure 3 vaccines-13-00988-f003:**
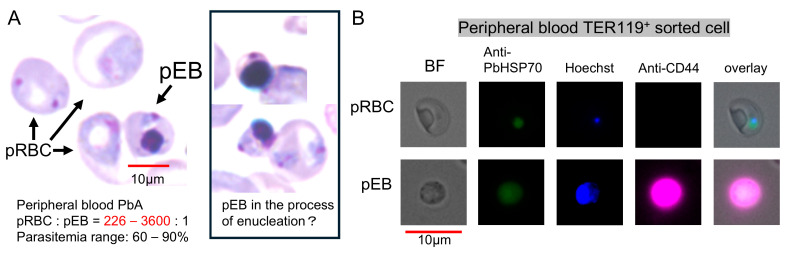
Detection of parasitized erythroblasts (pEBs) in the peripheral blood of mice immunized with *Plasmodium yoelii* 17XNL (PyNL) and subsequently challenged with *P. berghei* ANKA (PbA). (**A**) At approximately 20–30 days post-challenge, pEBs were more readily detected microscopically in Giemsa-stained peripheral blood smears compared with the primary PyNL infection. The ratio of PbA-parasitized red blood cells (pRBCs) to pEBs in PyNL-immunized and PbA-challenged mice ranged from 226–3600:1, with 60–90% parasitemia. In contrast, non-immune mice examined 7–10 days after PbA infection exhibited no detectable pEBs (PbA pRBCs:pEBs = countless:0; N > 100 mice, 0.5–20% parasitemia). Right panel: Representative pEB showing enucleation-like morphology. Scale bar: 10 µm. (**B**) TER119^+^ cells collected from immunized and challenged mice were stained for CD44 (surface markers) and for intracellular *P. berghei* HSP70 and Hoechst. Scale bar: 10 µm.

**Figure 4 vaccines-13-00988-f004:**
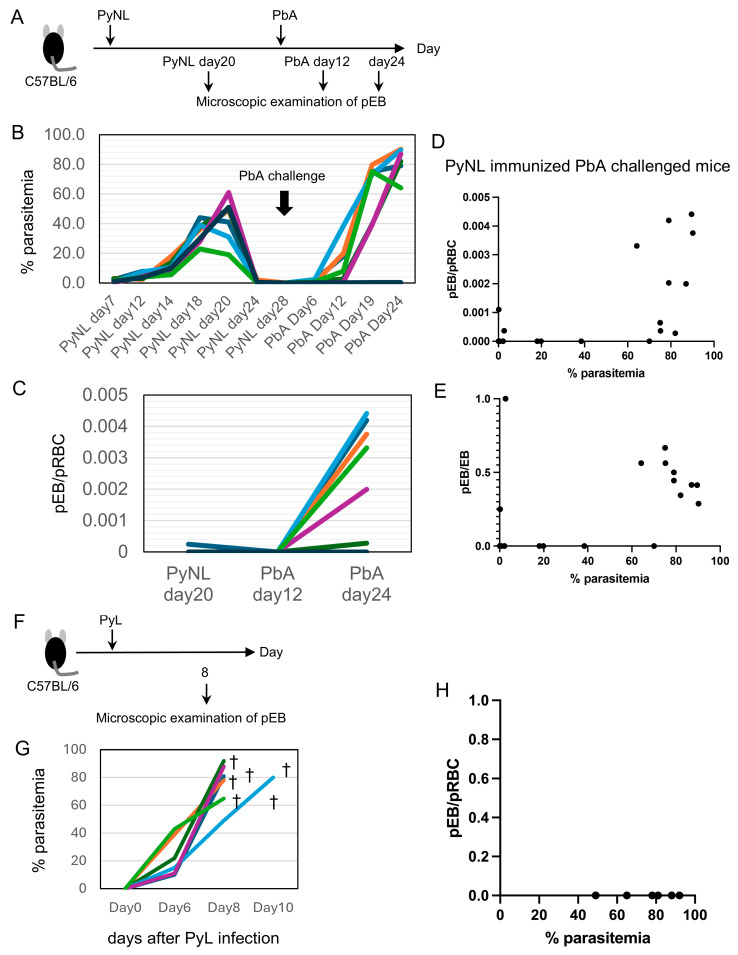
Increased proportion of parasitized erythroblasts (pEBs) in peripheral blood. (**A**) Experimental protocol for monitoring and increased proportion of pEBs in peripheral blood and the timing of pEB determination. (**B**) Time-course analysis of parasitemia from primary infection with *Plasmodium yoelii* 17XNL (PyNL) to challenge infection with *P. berghei* ANKA (PbA). PbA was injected 40 days after PyNL infection. Results from seven representative mice out of a total of 21 are shown. Each line with a different color represents data from an individual mouse. (**C**) Microscopic examination of pEBs in PyNL-infected mice on day 20 and in PbA-challenged mice on day 12 and day 24 post-infection (pEBs/pRBCs are shown). Results from seven representative mice out of a total of 21 are shown. Each line with a different color represents data from an individual mouse. (**D**,**E**) Microscopic analysis of pEBs following PyNL immunization and subsequent PbA challenge on day 12 and 24 ((**D**): pEBs/pRBCs vs. % parasitemia, (**E**): pEB/EB vs. % parasitemia). Each dot represents an individual mouse (N = 21 mice). (**F**) Protocol for microscopic examination of pEBs in mice infected with *P. yoelii* 17XL (PyL) at day 8 post-infection. (**G**) Parasitemia and survival of PyL-infected mice. † indicates death (N = 6 mice). Each line with a different color represents data from an individual mouse. (**H**) No pEBs were detected at day 8 in any PyL-infected mice after examining 10,000–40,000 pRBCs. N = 6 mice.

## Data Availability

Data are contained within the article.

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
