# Peer review of "Visual Detection of Malaria Parasite-Parasitized Erythroblasts in Peripheral Blood via Immunization-Based Model"

_vaccines, 2025, doi:10.3390/vaccines13090988_

Round 1
Reviewer 1 Report (New Reviewer)
Comments and Suggestions for Authors
Reviewer 1 would like to thank the authors for consdering the reviewer's feedback, for the point-by-point reply and for the revisions in the manuscript. The manuscript clearly improved by addressing reviewers' feedback.
Line 50: "in studying pEB is the rarity of peripheral blood" this sentence is incomplete, please revise to e.g. "in studying pEB is the rarity of these cells in peripheral blood" or similar.
Line 61: the authors are still referring to "laborious" maintenance of Plasmodium falciparum (Pf) cultures
Line 276: "On PbA day 12, no pEBs were detected." The sentence seems incomplete.
Line 278: "In contrast, in long-term surviving 278 mice that mounted strong immunity and maintained parasitemia below 1%, no pEBs 279 were observed." This seems to be in contrast to the initial hypothesis that pEBs play a role in e.g. protective immunity. Or is parasitemia too low to detect pEB? Please clarify.
Line 361: "While infection with PyNL affects erythroblast differentiation, particularly the maturation of RBC." - please correct, the sentence seems incomplete.
In the newly added text, typos and spelling errors require correction.
Author Response
Please see the attachment.

Reviewer 2 Report (Previous Reviewer 2)
Comments and Suggestions for Authors
The manuscript has been improved with the addition of new data.
Author Response
We are deeply grateful to the reviewers for their constructive feedback and for recognizing the value of our work.
Reviewer 3 Report (Previous Reviewer 3)
Comments and Suggestions for Authors
In the manuscript, the authors describe a novel approach for studying Plasmodium-infected erythroblasts, based on pretreatment of mice with live P. yoelii 17XNL followed by challenge with P. berghei ANKA. This study provides a valuable platform for in vivo/ex vivo analysis of pEB biology without the need for bone marrow or spleen isolation, thereby lowering both technical and ethical barriers in the field. The manuscript offers a detailed description of the methodology and the findings obtained. However, I would like to suggest that the authors revise a few aspects:
- In Figure 3, the CD44 fluorescence image of pEBs is overexposed (signal exceeds cell boundaries). Consider reducing staining intensity and/or contrast.
- Ensure correct use of italics for genus and species names.
- Lines 361–363 contain incomplete sentences; please revise for coherence.
Beyond my observations, the methodology presented here could improve diagnostic precision and advance our understanding of host–parasite interactions. It may also provide key insights into immune responses in contexts of exposure to multiple parasite species and guide the optimization of vaccine strategies and novel therapies.
Author Response
Please see the attachment.

Reviewer 4 Report (Previous Reviewer 4)
Comments and Suggestions for Authors
The current version of the manuscript satisfactorily reflects my suggested contributions and, therefore, I consider the article ready for publication.
Author Response
We are deeply grateful to the reviewers for their constructive feedback and for recognizing the value of our work.
This manuscript is a resubmission of an earlier submission. The following is a list of the peer review reports and author responses from that submission.
Round 1
Reviewer 1 Report
Comments and Suggestions for Authors
In the manuscript, the authors decsribe a novel model to study parasitized erythroblasts (pEBs) in vivo. pEBs numbers are increased in peripheral blood by immunizing mice with live Plasmodium yoelii 17XNL, followed by challenge with P. berghei ANKA. pEBs are assessed by microscopic examination for morphology and counts/ratios.
General concept comments:
The development of an in vivo model to study pEBs is of interest. However, pEB are mainly detected in the peripheral blood of animals with high parasitemia. Can authors rule out any artefact due to the immunization method and high parasitemia (e.g. above 60%)? What is the biological significance of parasitized erythroblasts if they cannot be (or rarely be) detected in natural infections?
Immunizations are performed with pRBCs that are injected intraperitoneally. Could authors provide rationale why PyNL and PbA pRBCs were injected intraperitoneally for immunizations? What is the route for challenge by pRBCs – intravenously or intraperitoneally?
Please provide a rational why maintenance of Plasmodium falciparum (Pf) cultures is considered laborious. It is a standard technique routinely used in many laboratories.
The authors state that P. yoelii is a model for P. vivax while P. berghei is considered a model for P. falciparum. While the named parasites possibly have similarities, e.g. P. falciparum and P. berghei have different genes and behave differently in pathogenesis (e.g. cytoadherence of PfRBC). Therefore, the claim seems incorrect.
The main message of the manuscript is not clearly outlined. While the development of the pEB model should be in focus, the overall message of the manuscript is unclear, and is moving from microscopy findings to vaccine approaches without clear link – only indicating that pEBs could play a role in vaccine-related immune responses. No data is shown here to support this.
No data is shown on the mechanisms for inducing the high pEB numbers in the model. No surface markers were used to confirm that detected cells are EBs.
Additionally, quantification of pEB by qPCR and/or flow cytometry or any other method than microscopy would have supported the microscopical findings.
The figures in the manuscript version shared with reviewers is of very low quality and figure D impossible to read.
Overall, the manuscript is not conveying a clear message, and method development, vaccinations, read outs are not consistently used to create a common theme.
Specific comments:
Line 93-95: "Parasitemia was assessed by blood smear fixed with methanol and stained with 3% Giemsa solution Sigma-Aldrich, St. Louis, MO, USA or Nakalai Tesque, Kyoto, Japan), counting at least 2,000 RBCs per sample 2 or 3 times per week."
Please explain how assessing parasitaemia 2 or 3 times per week was used to determine parasitaemia at the time of the mouse injections.
Line 98-101: How were PbA for challenge administered? Were they also administered intraperitoneally? What was the parasitaemia of the injected pRBCs? Only the overall count is indicated.
Line 103-104: "blood smears were randomly prepared from 17 specimens obtained from 21 PyNL-immunized, PbA-challenged mice" - please clarify what "randomly prepared from 17 specimens obtained from 21 mice" is referring to? What are randomly prepared specimens? Why are not all mice sampled and smears read at each of the timepoints?
Line 108-118: please restructure the method section for the Microscopic examination of pEB. The section also seems to provide inconsistent information. Did all eight researchers perform microscopic examination? Why is only quantitative data from three trained researchers used.
Figure 1C only shows survival rates of mice but from data provided, protection from cerebral malaria cannot be seen. The manuscript also mentions death by anaemia. Without any additional data, no reference should be made to the cause of death of the mice.
Comments on the Quality of English LanguageOverall, English language is fine, some typos would require correction.
Reviewer 2 Report
Comments and Suggestions for Authors
Ito et al. have developed a method to identify malaria parasite-infected erythroblasts following vaccination with live parasites. This report is particularly interesting as it focuses on an understudied life stage of malaria parasites. The experiments conducted were well-designed, performed, and analyzed.
Specific points
- The authors should provide the full name of the parasite lines and relevant references for each. For example, for "PyNL," they should specify "Plasmodium yoelii yoelii 17X NL" and cite its lineage (e.g., "original strain Landau and Killick-Kendrick, Trans R Soc Trop Med, 1960, 633; transferred to Cox, Bull WHO, 43, 325; then Playfair et al., Immunology, 1977, 32, 681; and subsequently to the authors' lab"). This detailed information is crucial for comparing studies and understanding the derivation of the strains.
- Line 126: The claim regarding neo-hematopoiesis in the spleen should be tempered given the recent increase in reports on this phenomenon.
- Since not all mice survived after PbA challenge, the authors should address the following: i) did these mice (that did not survive) have a high number of pEB (parasitized erythroblasts) in their blood? and ii) is this pEB population associated with protection against the challenge?
- Did the authors attempt to validate these cell populations using flow cytometry and cell sorting? This would be an important experiment to perform.
Reviewer 3 Report
Comments and Suggestions for Authors
The manuscript entitled “Visual Detection of Malaria Parasite-Parasitized Erythroblasts 2 in Peripheral Blood via Immunization-Based Model” describes a novel approach for studying Plasmodium-infected erythroblasts, based on pretreatment of mice with live P. yoelii 17XNL followed by challenge with P. berghei ANKA. The study provides a valuable platform for investigating erythroblasts in peripheral circulation. Nevertheless, several aspects could be refined to enhance clarity and facilitate the manuscript comprehension.
- In Section 2.1, the authors must state whether the animals were treated in accordance with bioethical standards and regulations for animal experimentation.
- In Section 2.2, it is recommended to specify that PyNL refers to the Plasmodium yoelii 17XNL strain and PbA to the berghei ANKA strain, as these designations are used for the strains throughout the manuscript.
Furthermore, it is recommended that the authors provide a detailed description of the treatment received by the animals, both in the control group and in the group pre-inoculated with PyNL and subsequently challenged with PbA.
- In Section 2.3, the authors should provide a more detailed account of how parasitemia was monitored and specify whether any particular analyses or observations were performed on the mice at the time of death.
- In Section 2.4, it is recommended that the authors either provide a more detailed description of the Giemsa staining procedure or cite an appropriate reference for the methodology used.
- In Section 2, the authors should provide a detailed description of the microscope and image-capturing equipment used, including the brand and supplier, as well as the optical magnification applied.
- It is recommended to clarify in Section 2.4 whether each investigator analyzed more than one preparation of each sample.
- In Section 3.1, “Visual Detection and Expansion of pEB in Peripheral Blood,” the authors should use caution or consider revising the term “expansion of pEB,” since although an increase in the proportion of pEBs in peripheral blood is observed in animals pretreated with the PyNL strain, this may not reflect increased production of these cells in the tissues of origin, but rather an enhanced release into circulation.
- The investigators analyzed this model in both male and female C57BL/6 mice; however, they do not report whether any sex-related differences were observed. Clarification of this point is recommended.
- In lines 131, it is recommended to modify the expression “PyNL-immunized mice served …” to “Immunization of mice with PyNL…” to improve the clarity and understanding of the sentence.
- In lines 132-133, the statement 'The mice did not develop experimental cerebral malaria (ECM) in the B6 strain...' does not clarify the meaning of “B6 strain.” Moreover, none of the results presented nor the methodology employed in this study analyze the development of cerebral malaria. If this refers to findings from previous studies, please provide the appropriate bibliographic reference.
- It is recommended to increase the resolution of the figures to enhance their clarity and visual quality.
- In Figure 1B, the authors are encouraged to additionally include a schematic diagram of the treatment regimen for the control group animals to enhance understanding of the treatment scheme for both experimental groups.
- In the legend of Figure 1F, it is recommended to clarify which experimental condition is represented by the graphed results.
- It is recommended that the legend of Figure 2 specify which images within the panel correspond to erythroblasts containing the parasite forms: merozoites, rings, trophozoites, schizonts, and gametocytes.
- The authors report that mice pretreated with PyNL fully recover from the infection by day 30, and that upon subsequent inoculation with PbA, the observed erythroblasts are infected exclusively with the PbA strain. However, no supporting data are provided for these claims. Clarification of this point is recommended.
- In lines 201–202, the statement “The ratio of PbA-parasitized polychromatic erythroblasts to orthochromatic erythroblasts in peripheral blood was 1:2 (Fig. 2)” is inaccurate, as Figure 2 does not depict this ratio and no quantification is provided. Please revise the wording of this sentence to improve clarity and accuracy.
- In the Discussion section, the authors should provide a more comprehensive explanation of the physiological relevance of detecting erythroblasts in the bloodstream, as well as the implications of infecting this erythropoietic precursor for disease progression.
- The Abbreviations section is currently missing; including it is recommended to enhance the clarity and readability of the text.
- The authors should update the bibliography.
Beyond my observations and suggestions, I consider that the publication of this manuscript offers a significant tool for investigating erythroblast infection by different Plasmodium species and provides a basis for exploring the impact of co-infection with diverse parasite strains on host erythropoiesis.
Reviewer 4 Report
Comments and Suggestions for Authors
I congratulate the authors for the originality of the study, which aimed to detect Plasmodium infected erythroblasts in peripheral blood using a murine immunization model with PpyNL followed by PbA. The study offers a novel in vivo and ex vivo system for investigating pEB Plasmodium interactions without the need for invasive procedures or genetically modified parasites.
The methodology was appropriately designed to meet the study's objectives.
I believe that the difficulty in identifying pEB(s) in peripheral blood through blood smears thus limiting the study of the function of these cells in Plasmodium infections as well as the technical limitations and the objective of the study , as described in ines 121 to 129, should be addressed in the introduction section. The objective is more clearly stated in the results section than im the introducrtion.
In the discussion, the relevant results were properly interpreted and compared with the literature in the research field. The conclusion is well´founded and aligns with the results.
These are the remarks I subnit.
